## [Transparent Peer Review file · Nature Communications]

A red/blue optoswitch for temporal control of chloroplast transcription and biogenesis in Arabidopsis

Corresponding Author: Professor Thomas Pfannschmidt

Version 0:

Reviewer comments:

Reviewer #1

(Remarks to the Author)

Overview

Uecker et al. present a method for controllable genetic complementation of Arabidopsis (PINC), which provides a means of analysing the functional role of essential plant proteins. They demonstrate the use of PINC to control assembly of the chloroplast transcription complex PEP via induction of one of its critical subunits, PAP7. Two major conclusions are drawn: (1) that chloroplast biogenesis is coupled to the developmental state of the cell it is in, and (2) assembly of the PEP complex does not depend on light or photosynthetic activity. The manuscript thereby presents significant advances in multiple areas through a straightforward and elegant experimental design.

The findings presented are all supported by the displayed data, and I did not identify anything that prevents publication. The methodology appears sound and reproducible, though this is not my area of expertise. However, there is substantial room to improve clarity in both the main text and the way figures are presented as detailed below.

Major comments:

- I can envisage PINC would be of broad utility to plant biologists as it does not depend on expensive technology. The manuscript focuses on the use of PINC to study proteins essential to chloroplast biogenesis. However, PINC would presumably be useful to study proteins that are essential for any reason, not only chloroplast biogenesis. Can the authors clarify in the text if this is the case, or if it is uniquely suited to the study of chloroplast biogenesis? If the PINC system is indeed of general utility, I would suggest adapting the abstract and summary to emphasise this at the outset and attract more readers than those interested in chloroplast transcription.
- Dose-dependency is an important feature of the system. Photosynthetic parameters show dose-dependency that suggest PAP7 levels increase with more BLRE repetitions, but the data on PAP7 expression levels (1c) do not show the same trend – seemingly because only three measurements were taken and one is unusually low in line BVB12. Repetition of this experiment to reduce uncertainty in BVB12 would give readers confidence that the system works in the way expected, and that they could choose a number of BLREs to have some control over the extent of expression (closer to 70% of wt with 1x or to 100%? of wt with 3x).
- The inability to completely complement to wt levels (evident in 1d-e) is an important limitation that could be more clearly quantified. The leaky expression in the 4xBLRE off condition led the authors to analyse plants with 3xBLREs. It would be helpful for readers if (1) photosynthetic efficiency (fig 1d) was quantified, and (2) the main text stated the attributes of the line selected as useful (BVB09): PAP7 expression relative to wt as a %, photosynthetic efficiency as a % relative to wt, total chlorophyll levels as a % relative to wt. Each of these numbers will likely reassure readers that although the complementation level is not 100% it is still relatively high.
- The conclusion there is a 'time window' (line 173) was not immediately obvious from the data. I strongly suggest the authors explain the reasoning behind this conclusion. In particular, are they confident from this experiment alone there is both a cell age that is too young and a cell age that is too old for PEP activation to be sufficient for greening? Or is it only the threshold of too old that can be observed?

- Regarding the time range (line 176): 'crucial' is a problematic choice of word as PEP is likely also crucial in other times. A better phrasing would be in terms of 'necessary' and 'sufficient': PEP is both necessary and sufficient in the first time period and is presumably still necessary but no longer sufficient in the second time period. Is that how the authors see it?
- The 'patchy' complementation seen in the three-day shift may arise because the cells at the centre of the leaf are younger and were not exposed to blue light for enough time for sufficient PAP7 accumulation to occur, therefore remaining white. Is this the authors interpretation too? Or do they propose that cells can be too young for their plastids to green? Could these alternatives be investigated by checking what the minimum duration of blue-light exposure needed for greening to occur is?
- The authors note that DCMU increases the abundance of PEP specifically in BVB09 and not wt. This appears striking: the difference appears to be 10-fold or more in the 40 uM treatment and exceeds PEP abundance in wt. Could the authors comment in the Discussion what this might indicate? Can it be concluded that an increased expression of PAP7 relative to wt leads to increase in the production of all PEP subunits? PAP7 expression quantification on DCMU-treated plants may provide the necessary insight.
- In line 266 there is a risk of overstating how surprising this is. The data presented does not indicate that PEP does not have a 'predominant role', only that another factor is also required and is lost with time. The major discovery appears less about the role of PEP, and more about the identification there is another essential factor in chloroplast biogenesis (of which there are many) that is no longer available.
- It is not introduced how this system differs from recent publications on plant optogenetics (refs 47, 48). This seems important context that could be provided in the introduction, rather than in the final paragraph. It is also not clear to a reader in what way the 'experimental goal and working principle' are different – additional detail would help emphasise the novelty of this work.

Minor comments:

- The authors seem to switch between PINC and BVB optoswitch to describe their technologies. Is it necessary to have both? Could PINC be used throughout?
- Line 32: 'show the effect of temporally...' would be clearer
- Line 34: 'allows reconstitution of PEP...' (preferable) or 'allows us to reconstitute PEP'
- Line 37: 'is effective only' might be changed to 'can only occur'
- Line 39: 'to grow' should be 'growth of'
- Line 66: 'photosynthesis' should be 'photosynthetic'
- Line 68: 'plastid-own genome' should be 'plastid genome' (preferable) or 'plastid's own genome'.
- Line 73: comma needed after 'ancestor'
- Line 77: 'cyanobacterial crab-like structure' should be 'bacterial crab claw architecture' or similar (the crab claw shape is not unique to cyanobacteria).
- Line 79: 'put forward' should be 'advanced' or similar
- Line 80: 'it does not provide direct clues' – I would argue the structure does provide clues (or hypotheses more specifically). What it does not provide is proof, which is what makes the described genetic analysis essential.
- Line 82: should end with 'in Arabidopsis' for clarity?
- Line 86: 'hardly produce' could be 'produce few' or 'generally do not produce' for clarity.
- Line 92: 'allow to grow' should be 'allow growth of' (preferable) or 'allow us to grow'.
- Line 100: 'As regulatory tools' should be 'As the regulatory element' or similar.
- Line 102: 'Basic' should be 'The basic'
- Line 112: 'are' should be 'were'
- Line 115: 'a point-of-no-return' is a somewhat unambiguous expression. Perhaps the authors would consider alternative wording, such as: 'that initiation of chloroplast biogenesis must occur before a specific timepoint during leaf development.'
- Line 117: 'potential photosynthetic influences' is ambiguous. Perhaps the authors would consider the phrasing: 'to test whether PEP formation depends on, or is influenced by, photosynthetic activity'
- Line 119: The authors need to clarify whether they are referring to PEP assembly or activity – what has been measured?
- Line 121: comma needed after 'optoswitch'.
- Line 128: 'PEP complementation' would be unclear to some readers. 'for controlled complementation of PEP biogenesis' may be clearer?
- Line 133: could a citation be provided to the publication(s) pap7 phenotype was originally characterised?
- Line 135: (BLRE)
- Line 144: 'effective' could be 'switchable/controllable'?
- Line 144: A significant improvement would be to clarify what each line is in the main text – currently the reader must look at the figure for this. For example: 'representative lines with 2 (BVB04), 3 (BVB09) or 4 (BVB12) BLRE repetitions developed...'
- Line 145: '1c' should be '1b' and '2c' should be '1c'
- Figure 1c: 'relative expression' could be more specific on the figure: 'Relative PAP7 expression'.
- Line 145: comma needed before 'with'
- Line 145: 'close to' could be made more precise with a percentage: 'approximately 70-90% of wild type levels'?
- Line 149: 'photosynthesis' should be 'photosynthetic'
- Line 149-150: commas needed: ', especially in older plants, '
- Line 157: could be clarified – this is presumably multiple individual plants, not one plant being shifted at different time points? The word 'it' suggests an individual. 'plants were grown in... before being shifted to..' would be clearer. Figure 3a

legend would likewise benefit from clarifying these were different seedlings transferred to blue light at different time points, as it would be possible a reader would think this is the same seedling. Could the wording be 'developmental timeline showing representative images of albino ...' Or similar?

- Line 158: Could be stated more clearly that the conditional is simply enough time for follow up leaves to grow: perhaps 'and albino follow-up leaves if cultivated in Roff for long enough for these to develop'?
- Line 165: comma needed before 'where'. This statement would benefit from a citation.
- Line 170: comma needed before 'perfect'
- Line 172: 'i)' should be before 'demonstrate' (as 'suggest' is after ii)
- Line 173: 'time window' may not be the best way to describe as there is a limit to the latest point but not the earliest – see comment above.
- Line 177: 'and performed' could be 'by performing'
- Line 183: 'biogenic' is not clear to this reviewer. If this has a specific meaning, could the authors clarify in the text what is meant? If not, this word could be removed.
- Line 191: comma needed before 'suggesting'
- Line 194: Heading should be clearer, and would benefit from being in the form of a conclusion: 'Production of PEP does not depend on photosynthesis' for example.
- Line 196: comma needed before affecting
- Line 197: 'the perfect tool' appears an overstatement: perhaps 'provides a tool'?
- Lines 219-220: commas needed ', and also that of ... RNAP II, '
- Line 229: 'especially' – do the authors mean 'notably', or that PRIN2 has particularly little responsiveness to DCMU?
- Line 235: comma needed before 'suggesting'
- Line 236: Should be 'In the case'
- Line 239: 'inhibitor' should be 'inhibitory'
- Line 245: comma needed before 'which'
- Line 247: 'tailor-made' is not needed
- Line 248: 'alternatively' should be 'additionally'?
- Line 249: comma needed before 'allowing'
- Line 251: 'highly' should be 'greatly' or 'substantially'
- Line 253: comma needed before 'which'
- Line 259: comma needed before 'indicating'
- Line 261: should be 'complicated'
- Line 265: comma needed before 'indicating'
- Line 266: a citation is needed for what the 'current models' are.
- Line 271: 'identical' appears an overstatement: to what level of precision? Perhaps 'consistent with the timescale of retrograde..'?
- Line 272: should include 'in wheat' to clarify it is not necessarily directly comparable.
- Line 272: should be 'The physical nature'
- Line 277: comma needed before 'indicating'
- Line 279: comma needed before 'function'
- Line 288: should be either 'PAP6, PAP10...' or 'PAPs 6, 10,..'
- Line 290: comma needed before 'regulation'
- Line 300: comma needed before 'our'
- Line 311: could this be 'in standard growth cabinets equipped with...' for simplicity?
- Line 314: why must the phenotype be visible/trackable? Molecular techniques could be used too surely?
- Figure 1e: there appear to be too few significance comparison on the chart. Could differences between groups be presented as letters using a suitable post-hoc test?
- Figure 2 is labelled Figure 1.
- Figure 1d and Figure 2d would benefit from quantification (bar charts).
- Figure 6: the colouring and key are not clear. The key for 'shift' should at least include red/blue colours behind the lines. However, I strongly suggest having its own colour, rather than attempt to convey experimental approach inside a bar chart that may be mistaken to be a divided bar chart where two measurements are shown within a column.

Reviewer #2

(Remarks to the Author)

This study from the Pfannschmidt laboratory generates an elegant, simple yet ingenious optogenetic system for use in plants. Several optogenetic systems have been developed and refined over the last two decades. Their aim is to control gene expression in an on/off manner, ideally fulfilling a number of requirements: (1) no need for exogenous chromophores, (2) minimal interference with endogenous signalling pathways, (3) compatibility with the need for light cycles for plant growth and (4) be widely applicable.

The system described by Uecker et al. beautifully fulfils requirements 1 and 3. 2 and 4 are achieved with some limitations. As the data themselves show, monochromatic red light used for growth results in plants in which the lack of blue photoreceptor action causes, among visible phenotypes, elongated petioles and leaf blades which are unable to face the light sources. The applicability of the system will depend on the ability of the blue light-responsive element of the CHS gene, identified in *Arabidopsis thaliana*, to be active in other species.

While those limitations cannot be questioned, use of the system in *Arabidopsis* is sufficient to justify its utility. Furthermore the authors develop it for one specific purpose: to generate genetic stocks of mutants whose homozygosity is either lethal or, at least, incompatible with seed production, and therefore with propagation. This then allows refined experiments with

homozygous material to identify the consequences of presence/absence of gene function, repeated multiple times: the relevant, homozygous stock is generated under standard growth (white light) conditions, and the control achieved by turning complementation off by transfer to monochromatic red (i.e. blue-less) light. This is entirely, beautifully achieved, applying the technique to study the function of the plastid-encoded RNA polymerase (PEP), through its requirement of nucleus-encoded PAP7/pTAC14. The study demonstrates the usefulness of the technique, and reports three key findings: the requirement for PEP function during a critical time window in cells in which early stages of chloroplast development are taking place, but very little later on, the fact that this requirement is (as expected) cell-autonomous, and lastly that PEP activity appears to be minimally dependent of photosynthetic function (photosynthetic electron transport) during these biogenesis stages. The unexpected result of the activity of PEP under photosynthesis-inhibited conditions is precisely described by monitoring both transcript levels and overall protein composition of the relevant, dissected tissues.

One issue may require further evidence to confirm the stated critical window and “point of no return” of PEP requirement. The evidence provided shows that seedlings grown for up to three days in red light (R), or in darkness, can have PAP7 activated by switch to blue (B) and the cotyledons green, but this is not the case after four days (of R or darkness). The authors conclude that PEP activation later cannot cause chloroplast greening. This interpretation would require demonstration that PEP activation by B did indeed take place after 4 days in R or darkness. The endogenous CHS activation system used might, conceivably, also have a window of competence, and this would explain the same observation. One easy test would be to monitor the transcript of the transgene, making use of the fused GFP, as done in Fig. 6.

The reviewer also found the cell-autonomy of the above response particularly intriguing. Thresholds of activity of essential, cell-autonomous processes in chloroplast biogenesis clearly exist, are probably operating in early proliferating cells in leaf primordia, and explain the development of cell sectors of contrasting phenotype, as seen in variegated leaves. The evidence for this cell-autonomous requirement is presented in Fig. 3 and shown by what seems a cotyledon (Fig. 3B, possibly derived from 3A). Cotyledons are, of course, embryonic organs. Was such a phenomenon ever observed in newly-developing leaves emerging from the meristem? A seedling in Fig. 3F may show this, but the image is insufficient to confirm. It would be of value if the authors could clarify.

Lastly, the authors display how transcript levels of a range of genes, nucleus- and plastid-encoded, are impacted by the absence of PAP7 or its complementation. Among plastid-encoded genes, those transcribed by the nucleus-encoded polymerase are (with the exception of *rpoB*) all presented in Fig. S4. Do the authors have an explanation for the low transcript levels of *petD* in the *pap7* mutant and the R-grown BVB line?

The text is generally well written, and the figures beautifully constructed to easily interpret. Note, however, that Fig. 2 is also labelled as a second Fig. 1. On several occasions Fig. 2 is referred to as Fig. 1, and viceversa.

Editing comments:

Repetitions > repeats.

Line 79 put forward > advanced (to “put forward” is done only the first time)

144 Should refer to Fig. 1B-C and Fig. S2C (lines BVB04m BVB09 and BVB12).

154 No Fig. 1f exists. Fig. 2f does not show leaky expression of 4-repeat BVB

Fig. 6 What is given a value of 1, y axis?

261 Complicate > complex

Fig. 1 legend Ron > Bon

Methods should indicate the statistical tests used. Note that multiple testing increases errors, and there are ways to avoid those.

Reviewer #3

(Remarks to the Author)

The manuscript by Uecker et al. addresses an important and timely question: the role of plastid-encoded RNA polymerase (PEP) in chloroplast biogenesis. Recent progress in structural understanding of PEP has only reinforced the importance of open questions regarding the biological roles of this crucial enzyme. Advancement in this field has been hindered by a challenging chicken-and-egg situation, where disruptions of PEP subunits impede chloroplast biogenesis, subsequently disrupting various processes, potentially including transcription itself. This manuscript addresses this conundrum by developing a novel inducible system, enabling precise control of PEP subunits (or any other proteins) through exposure to different wavelengths of light. Utilizing this tool, the manuscript investigates the role of PEP and elucidates two important aspects of its involvement in chloroplast biogenesis. (1) A limited time window for PEP-dependent chloroplast development, which is an interesting novel finding and (2) independence of chloroplast development of PEP transcription. Although these insights do not exhaust the power of the new inducible system, the current scope of the manuscript is appropriate and fully justified by the impact of the novel optogenetic system.

This is a significant study that will establish a new standard for investigations of PEP subunits, other essential chloroplast proteins, and likely other essential plant proteins. The manuscript is well-written, technically solid and will be of interest to a broad audience of scientists studying genetics, not only in plants.

Specific comments

- There is no data showing behavior of the BVB optoswitch on the level of RNA or protein accumulation. It is important to know the level of induction and the level of leaky expression, which is a common problem of many inducible systems. Adding new experimental data providing these insights would substantially strengthen the manuscript.
- The mechanism responsible for the checkpoint that provides competence for chloroplast biogenesis should be discussed more thoroughly. For instance, is it possible that this checkpoint is mediated by nutrient availability in seeds?
- There is no justification of why the new optogenetic system is better than previously established inducible systems in plants. There is also no comparison to other optogenetic inducible systems.
- There is no discussion of potential impacts of growing plants in narrow spectra (red or blue light only) compared to white light, which is an important consideration for the interpretation of results obtained using the BVB optoswitch.

- DCMU treatment has been used in the past and citing and discussing original literature describing its limited impacts on chloroplast biogenesis would strengthen the conclusions of this manuscript.
- Impacts of DCMU on RNA accumulation are sometimes overinterpreted as impacts on gene expression. This part of the manuscript has limited influence on its conclusions and could easily be shortened.
- There are problems with labels of Fig 1F (referenced in the manuscript) and Fig 2 (mislabeled in figure legends).

Version 1:

Reviewer comments:

Reviewer #1

(Remarks to the Author)

All comments were appropriately addressed, and I see no barrier to publication in its current form. Congratulations to the authors on this important contribution.

Reviewer #2

(Remarks to the Author)

This manuscript, already of high quality in its first version, has been significantly strengthened by the additional data, specially on PEP protein accumulation and function and on functionality of the blue-dependent promoter element, the improved figures, and the substantially revised text. The efficiency of the optoswitch in particular is robustly confirmed. The authors refer to their responses (as is customary) as a rebuttal letter, but by-and-large they have taken the majority of the reviewers' comments on board. The study does, without question, contribute new understanding of chloroplast development and provide a useful, promising research and biotechnological tool.

Minor corrections:

Line 405: Please replace "pretends" with "mimics"

Line 443 (and possibly elsewhere): Replace "a well known phytochrome response" with "a result to absence of cryptochrome and phototropin photoreceptors signalling", or an equivalent expression. The current statement is incorrect. Phytochrome mutants, some in particular exhibit elongated petioles, therefore phytochrome signalling would shorten the petioles, and shade-light, therefore REDUCED phytochrome signalling, elongates them. The elongated petioles are a result of low cryptochrome action, and the down-turned or twisted laminae a consequence of absence of phototropin signalling.

Reviewer #3

(Remarks to the Author)

I am satisfied with the revisions and support publication of the revised manuscript.

Point-by-point response to reviewers' comments

We went carefully through all comments and improved manuscript, figures and supplements accordingly either by adding experimental data and/or by revising the respective text. Changes in the manuscript text are indicated by track changes function and are further detailed below. To facilitate the overview of changes for the reviewers we highlighted the principle changes in green.

Reviewer #1 (Remarks to the Author):

Overview

Uecker et al. present a method for controllable genetic complementation of Arabidopsis (PINC), which provides a means of analysing the functional role of essential plant proteins. They demonstrate the use of PINC to control assembly of the chloroplast transcription complex PEP via induction of one of its critical subunits, PAP7. Two major conclusions are drawn: (1) that chloroplast biogenesis is coupled to the developmental state of the cell it is in, and (2) assembly of the PEP complex does not depend on light or photosynthetic activity. The manuscript thereby presents significant advances in multiple areas through a straightforward and elegant experimental design.

The findings presented are all supported by the displayed data, and I did not identify anything that prevents publication. The methodology appears sound and reproducible, though this is not my area of expertise. However, there is substantial room to improve clarity in both the main text and the way figures are presented as detailed below.

Major comments:

- I can envisage PINC would be of broad utility to plant biologists as it does not depend on expensive technology. The manuscript focuses on the use of PINC to study proteins essential to chloroplast biogenesis. However, PINC would presumably be useful to study proteins that are essential for any reason, not only chloroplast biogenesis. Can the authors clarify in the text if this is the case, or if it is uniquely suited to the study of chloroplast biogenesis? If the PINC system is indeed of general utility, I would suggest adapting the abstract and summary to emphasise this at the outset and attract more readers than those interested in chloroplast transcription.

Response: We totally agree with this comment. The basic principle of the PINC strategy is suitable for any kind of essential protein in plants and is NOT restricted to chloroplast transcription and biogenesis although the phenotypic effects are here especially visible. We adapted the title and improved the abstract accordingly, we further recommend the system for general utility at the end of introduction and discussion. By choosing the right promoter elements the strategy might be also suitable even for other model organisms such as bacteria, yeast or even animals (worms, flies etc.). Without experimental data this might be however to speculative for discussion.

- Dose-dependency is an important feature of the system. Photosynthetic parameters show dose-dependency that suggest PAP7 levels increase with more BLRE repetitions, but the data on PAP7 expression levels (1c) do not show the same trend – seemingly because only three measurements were taken and one is unusually low in line BVB12. Repetition of this experiment to reduce uncertainty in BVB12 would give readers confidence that the system works in the way expected, and that they could choose a number of BLREs to have some control over the extent of expression (closer to 70% of wt with 1x or to 100%? of wt with 3x).

Response: The reviewer is absolutely right, dose-dependency is indeed an important feature of the BVB optoswitch. It becomes, however, more apparent with increasing plant age. While expression levels of PAP7 in cotyledons of BVB04, BVB09 and BVB12 (Fig. 1B) do exhibit induction under B_{on} conditions with relatively similar overlapping values (Fig. 1c), the phenotypic consequences on photosynthetic capacity and pigment content in later stages show a clear dose dependency (Fig. 1e,g,h). We added additional data showing phenotypes of BVB04, BVB09 and BVB12 in later stages (with approx. 8-10 leaves) alongside with the respective PAP7 expression in these plants (new Fig. 1d,e). Here, the dose dependency of the gene construct expression becomes clearly visible (with 50-60% of wt for 2xBLRE; 75% of wt for 3xBLRE and 85-95% of wt for 4xBLRE). This corresponds perfectly to the phenotypic observations at that developmental stage (which further corresponds to the data shown in former Fig. 1d,e now Fig. 1f,g,h). The figure was expanded accordingly and the text was adapted indicating the dose dependency in more detail.

- The inability to completely complement to wt levels (evident in 1d-e) is an important limitation that could be more clearly quantified. The leaky expression in the 4xBLRE off condition led the authors to analyse plants with 3xBLREs. It would be helpful for readers if (1) photosynthetic efficiency (fig 1d) was quantified, and (2) the main text stated the attributes of the line selected as useful (BVB09): PAP7 expression relative to wt as a %, photosynthetic efficiency as a % relative to wt, total chlorophyll levels as a % relative to wt. Each of these numbers will likely reassure readers that although the complementation level is not 100% it is still relatively high.

Response: The reviewer raises a highly important point that was very difficult for us to decide on. Either we could chose to go for a complete complementation but with some leakiness in the R_{off} condition or we could go for non-complete complementation but with full tightness in R_{off} . We chose the latter condition to avoid any ambiguities for further experiments since we were convinced that this would provide more conclusive experimental data on the role of PEP in chloroplast biogenesis and its initiation. We now provide the requested quantification data and, in addition, explain why the tightness of the system is more important than a full complementation.

- The conclusion there is a 'time window' (line 173) was not immediately obvious from the data. I strongly suggest the authors explain the reasoning behind this conclusion. In particular, are they confident from this experiment alone there is both a cell age that is too young and a cell age that is too old for PEP activation to be sufficient for greening? Or is it only the threshold of too old that can be observed?

Response: In the last two years we made many different light switch experiments with varying time ranges in R_{off} and B_{on} conditions. In the beginning of our studies we were just happy to get green leaves, but later we realized that fully developed albino leaves never started to green, although they got the light induction. This led us to conclude that there is a sensitive phase in leaf cells in which B_{on} condition are able to induce chloroplast biogenesis. Our further studies than demonstrated that cells that exceed a certain age do not green anymore suggesting an age threshold that, if passed, does not allow chloroplast biogenesis anymore. In sum our data indicate that there is a threshold of too old. We removed the term „time window“ and rephrased the text to make this more clear.

- Regarding the time range (line 176): 'crucial' is a problematic choice of word as PEP is likely also crucial in other times. A better phrasing would be in terms of 'necessary' and 'sufficient': PEP is both necessary and sufficient in the first time period and is presumably still necessary but no longer sufficient in the second time period. Is that how the authors see it?

Response: The reviewer is right, PEP is crucial not only in early phases but also in later stages. We rephrased this sentence to avoid any misunderstandings.

- The 'patchy' complementation seen in the three-day shift may arise because the cells at the centre of the leaf are younger and were not exposed to blue light for enough time for sufficient PAP7 accumulation to occur, therefore remaining white. Is this the authors interpretation too? Or do they propose that cells can be too young for their plastids to green? Could these alternatives be investigated by checking what the minimum duration of blue-light exposure needed for greening to occur is?

Response: We think it works the other way around, the cells get enough blue light but the white cells are already too old. The figure is a bit tricky in this respect, but we had no idea for a figure outline that is better than the present one. If the reviewer has any suggestion to improve it, we would be happy to change the figure accordingly. In fact it works like this: Seeds are sown in parallel on several Petri dishes that obtain 1 or 2 or 3 or 4 or more days of red light. Then they are shifted for 10 days to blue light. Thus they all obtain enough time in blue light to induce PAP7 even in cells in the center of the leaf blade. Seedlings that have obtained 4 days or more of red light before the shift to blue light did not generate green cotyledons anymore. Only subsequent leaves could green. Seedlings getting only 1 or 2 days red light after sowing, however, generated green cotyledons in the blue light and those getting 3 days in red light developed intermediate patterns with patchy leaves. We think the white cells in these leaves are too old to induce chloroplast biogenesis, while the green cells are younger (due to later cell division because of the special leaf blade development programme in dicot plants) and still are susceptible to the blue light induction. Since the duration of blue light illumination is not limiting we conclude that the age of the cells is the critical parameter. This is what we call the point-of-no-return. Once this point is passed by a cell it cannot generate chloroplasts anymore even if it gets enough blue light. In the case of the follow-up leaves in Fig. 2 the scenario is a bit different. Here tip and edges become green because these cells were developing in the leaf primordia in Bon. Then the shift to Roff was done and all cells that developed subsequently could not induce CP biogenesis. After a second switch to Bon these cells were however already too old for induction. We think that this is due to a predetermined developmental cascade that couples cell age with chloroplast development and that becomes desynchronized if PAP7 is not expressed in early stages of cell development. This is in full agreement with studies from Loudya et al who analysed different cell types along the developmental gradient in monocots. We improved the text in results and discussion accordingly and explain this now in more detail.

- The authors note that DCMU increases the abundance of PEP specifically in BVB09 and not wt. This appears striking: the difference appears to be 10-fold or more in the 40 uM treatment and exceeds PEP abundance in wt. Could the authors comment in the Discussion what this might indicate? Can it be concluded that an increased expression of PAP7 relative to wt leads to increase in the production of all PEP subunits? PAP7 expression quantification on DCMU-treated plants may provide the necessary insight.

Response: Yes, this observation is somewhat surprising. Apparently, DCMU treatment (actually done before the greening) promotes the accumulation of PEP complexes in BVB09, at least as detectable by the BN PAGE technique. We dont want to speculate too much on that since reviewer 3 is skeptical about DCMU effects on gene expression. However, the increased transcript accumulation of rpoB and GFP under these conditions (see Fig. 6AB) suggest that the strong inhibition of all newly emerging photosystem II complexes by the DCMU treatment pretends a metabolic or developmental plastid stage that is similar to an initial step of chloroplast biogenesis when NEP promotes PEP accumulation. The effect might be induced indirectly, e.g. by lack of metabolites such as sugars or

lipids. Further, it is possible that this effect is transient. The plastids in BVB09 are less well developed than those in WT, because their development started with 7 days delay (after the shift to Bon). Possibly NEP is more active in BVB09 than in WT in that moment. We **introduced these comments in the discussion as requested.**

Additional comment: The major determinant for PEP accumulation is likely the generation of RPO subunits as these build the core to which the PAPs associate. We regard it as unlikely that enhanced expression of a single nuclear-encoded PAP at transcription (as suggested by the reviewer) or translation levels governs the expression of RPOs. Currently, we assume that PAPs are produced in slight excess and attach to the RPO core in a stoichiometric number. We acknowledge the suggestion of the reviewer, but detection of PAP7 transcripts will not provide a sufficient explanation. Actually, the expression balance of PAPs is very delicate. Many experiments in our lab indicate that both, too low as well as exceeding expression of PAPs lead to pale phenotypes, possibly because this disturbs the number of PAPs that can be imported into plastids, for instance by cytosolic sequestration. We cannot explain this phenomenon right now since this requires precise quantification experiments which represent a project of its own. We aim to start a study on this, a project proposal is currently in preparation.

- In line 266 there is a risk of overstating how surprising this is. The data presented does not indicate that PEP does not have a 'predominant role', only that another factor is also required and is lost with time. The major discovery appears less about the role of PEP, and more about the identification there is another essential factor in chloroplast biogenesis (of which there are many) that is no longer available.

Response: We agree, that this point is a question of interpretation and can be understood also in a different way than we did. **We rephrased this section in a way that is less strict leaving it more open and mentioning the possibility of a yet unidentified factor.**

- It is not introduced how this system differs from recent publications on plant optogenetics (refs 47, 48). This seems important context that could be provided in the introduction, rather than in the final paragraph. It is also not clear to a reader in what way the 'experimental goal and working principle' are different – additional detail would help emphasise the novelty of this work.

Response: We were reluctant in introducing too many technical descriptions in the introduction as this would render our study into a more technological report which was not the aim of our project. However, we understand the concern of the reviewer that the progress of our PINC approach does not become apparent without more detailed knowledge about the current state of other approaches. We, thus, **introduced a paragraph in the introduction mentioning optogenetics and chemical induction systems which should help to place our system into the current technological context.**

Minor comments:

- The authors seem to switch between PINC and BVB optoswitch to describe their technologies. Is it necessary to have both? Could PINC be used throughout?

Response: PINC is the general approach that can be technically modified in many ways (e.g. by choosing different promoters or light wavelengths). BVB is the tool (with BVB+number to indicate the different genetic lines) that we have specifically developed for our study and scientific question. So

we think, both terms have their justification. **We gave additional explanation in the introduction and results sections to clarify this and adapted the terms where necessary.**

Response to the improvements given below: We thank the reviewer for this really careful consideration of our manuscript. **We introduced all the changes suggested below or performed modifications of the text that align with the reviewer comments (see tracked changes).** Below, we give only a few specific responses in cases where it seemed to us necessary for explanation.

- Line 32: 'show the effect of temporally...' would be clearer
- Line 34: 'allows reconstitution of PEP...' (preferable) or 'allows us to reconstitute PEP'
- Line 37: 'is effective only' might be changed to 'can only occur'
- Line 39: 'to grow' should be 'growth of'
- Line 66: 'photosynthesis' should be 'photosynthetic'
- Line 68: 'plastid-own genome' should be 'plastid genome' (preferable) or 'plastid's own genome'.
- Line 73: comma needed after 'ancestor'
- Line 77: 'cyanobacterial crab-like structure' should be 'bacterial crab claw architecture' or similar (the crab claw shape is not unique to cyanobacteria).
- Line 79: 'put forward' should be 'advanced' or similar
- Line 80: 'it does not provide direct clues' – I would argue the structure does provide clues (or hypotheses more specifically). What it does not provide is proof, which is what makes the described genetic analysis essential.
- Line 82: should end with 'in Arabidopsis' for clarity?
- Line 86: 'hardly produce' could be 'produce few' or 'generally do not produce' for clarity.
- Line 92: 'allow to grow' should be 'allow growth of' (preferable) or 'allow us to grow'.
- Line 100: 'As regulatory tools' should be 'As the regulatory element' or similar.
- Line 102: 'Basic' should be 'The basic'
- Line 112: 'are' should be 'were'
- Line 115: 'a point-of-no-return' is a somewhat unambiguous expression. Perhaps the authors would consider alternative wording, such as: 'that initiation of chloroplast biogenesis must occur before a specific timepoint during leaf development.'
- Line 117: 'potential photosynthetic influences' is ambiguous. Perhaps the authors would consider the phrasing: 'to test whether PEP formation depends on, or is influenced by, photosynthetic activity'
- Line 119: The authors need to clarify whether they are referring to PEP assembly or activity – what has been measured?
- Line 121: comma needed after 'optoswitch'.
- Line 128: 'PEP complementation' would be unclear to some readers. 'for controlled complementation of PEP biogenesis' may be clearer?
- Line 133: could a citation be provided to the publication(s) pap7 phenotype was originally characterised?
- Line 135: (BLRE)
- Line 144: 'effective' could be 'switchable/controllable'?

Rephrased into: While transformation of constructs with a single BLRE cassette did not yield green plants (Supplementary Fig. 2b), experiments done with constructs containing two, three or four BLRE repeats were successful.

- Line 144: A significant improvement would be to clarify what each line is in the main text – currently the reader must look at the figure for this. For example: 'representative lines with 2 (BVB04), 3 (BVB09) or 4 (BVB12) BLRE repetitions developed...'
- Line 145: '1c' should be '1b' and '2c' should be '1c'

- Figure 1c: 'relative expression' could be more specific on the figure: 'Relative PAP7 expression'.
- Line 145: comma needed before 'with'
- Line 145: 'close to' could be made more precise with a percentage: 'approximately 70-90% of wild type levels'?
- Line 149: 'photosynthesis' should be 'photosynthetic'
- Line 149-150: commas needed: ', especially in older plants, '
- Line 157: could be clarified – this is presumably multiple individual plants, not one plant being shifted at different time points? The word 'it' suggests an individual. 'plants were grown in... before being shifted to..' would be clearer.

Response: Yes, we did these experiments with many seeds/plants and show only one representative for each condition. The plants depicted in the photographs are indeed different plants. In Fig2 f,g however we show one individual plant each to demonstrate the developmental timeline of a single plant in response to repeated light shifts. We improved text and figure legend accordingly to make this clear.

Figure 3a legend would likewise benefit from clarifying these were different seedlings transferred to blue light at different time points, as it would be possible a reader would think this is the same seedling. Could the wording be 'developmental timeline showing representative images of albino' Or similar?

Response: Yes, each photograph depicts a different seedling since these are placed under blue light after the indicated time in red light. We improved the figure legend accordingly.

- Line 158: Could be stated more clearly that the conditional is simply enough time for follow up leaves to grow: perhaps 'and albino follow-up leaves if cultivated in Roff for long enough for these to develop'?
- Line 165: comma needed before 'where'. This statement would benefit from a citation.
- Line 170: comma needed before 'perfect'
- Line 172: 'i)' should be before 'demonstrate' (as 'suggest' is after ii)
- Line 173: 'time window' may not be the best way to describe as there is a limit to the latest point but not the earliest – see comment above.

Response: We rephrased this sentence.

- Line 177: 'and performed' could be 'by performing'
- Line 183: 'biogenic' is not clear to this reviewer. If this has a specific meaning, could the authors clarify in the text what is meant? If not, this word could be removed.
- Line 191: comma needed before 'suggesting'
- Line 194: Heading should be clearer, and would benefit from being in the form of a conclusion: 'Production of PEP does not depend on photosynthesis' for example.
- Line 196: comma needed before affecting
- Line 197: 'the perfect tool' appears an overstatement: perhaps 'provides a tool'?

Rephrased into: provides a suitable tool

- Lines 219-220: commas needed ', and also that of ... RNAP II, '
- Line 229: 'especially' – do the authors mean 'notably', or that PRIN2 has particularly little responsiveness to DCMU?
- Line 235: comma needed before 'suggesting'
- Line 236: Should be 'In the case'
- Line 239: 'inhibitor' should be 'inhibitory'

- Line 245: comma needed before 'which'
- Line 247: 'tailor-made' is not needed
- Line 248: 'alternatively' should be 'additionally'?
- Line 249: comma needed before 'allowing'
- Line 251: 'highly' should be 'greatly' or 'substantially'
- Line 253: comma needed before 'which'
- Line 259: comma needed before 'indicating'
- Line 261: should be 'complicated'
- Line 265: comma needed before 'indicating'
- Line 266: a citation is needed for what the 'current models' are.
- Line 271: 'identical' appears an overstatement: to what level of precision? Perhaps 'consistent with the timescale of retrograde..'?

Rephrased into: corresponds

- Line 272: should include 'in wheat' to clarify it is not necessarily directly comparable.
- Line 272: should be 'The physical nature'
- Line 277: comma needed before 'indicating'
- Line 279: comma needed before 'function'
- Line 288: should be either 'PAP6, PAP10...' or 'PAPs 6, 10,..'
- Line 290: comma needed before 'regulation'
- Line 300: comma needed before 'our'
- Line 311: could this be 'in standard growth cabinets equipped with...' for simplicity?
- Line 314: why must the phenotype be visible/trackable? Molecular techniques could be used too surely?
- Figure 1e: there appear to be too few significance comparison on the chart. Could differences between groups be presented as letters using a suitable post-hoc test?

Response: For clarity reasons we gave only those significance comparisons with immediate relevance to the parameter in question. In the Roff condition only the comparison between BVB12 and pap7-1 is important (because it indicates the potential leakiness) while comparisons between BVB04, BVB09 and pap7-1 were not significant. For a comparison to the wt control a statistical test was not required because of the eminent data differences. In the Bon condition only the differences between BVB04 with wt were significant, while those of BVB09 and BVB12 were not (indicating the dose dependency). For a comparison to the pap7-1 control a statistical test was not required again because of the eminent data differences. **We now added additional information about the non-significance in the differences and a comment in the figure legend where we indicate this including the statistical test used.**

- Figure 2 is labelled Figure 1.

Corrected.

- Figure 1d and Figure 2d would benefit from quantification (bar charts).

Introduced.

- Figure 6: the colouring and key are not clear. The key for 'shift' should at least include red/blue colours behind the lines. However, I strongly suggest having its own colour, rather than attempt to convey experimental approach inside a bar chart that may be mistaken to be a divided bar chart where two measurements are shown within a column.

Response: We agree, the colouring of the bars and the key for the shifted samples were neither consistent nor intuitive. Using a different colour would, however, destroy to some extent the style of our figures in our manuscript in which we always tried to support the understanding of the data by the colouring. We therefore decided to label the bars by a perpendicular red/blue colouring that suggests the red/blue switch during the growth period of the seedlings, but avoids the potential misunderstanding of a divided bar chart with two additive measurements. The key was adapted accordingly.

Reviewer #2 (Remarks to the Author):

This study from the Pfannschmidt laboratory generates an elegant, simple yet ingenious optogenetic system for use in plants. Several optogenetic systems have been developed and refined over the last two decades. Their aim is to control gene expression in an on/off manner, ideally fulfilling a number of requirements: (1) no need for exogenous chromophores, (2) minimal interference with endogenous signalling pathways, (3) compatibility with the need for light cycles for plant growth and (4) be widely applicable.

The system described by Uecker et al. beautifully fulfils requirements 1 and 3. 2 and 4 are achieved with some limitations. As the data themselves show, monochromatic red light used for growth results in plants in which the lack of blue photoreceptor action causes, among visible phenotypes, elongated petioles and leaf blades which are unable to face the light sources. The applicability of the system will depend on the ability of the blue light-responsive element of the CHS gene, identified in *Arabidopsis thaliana*, to be active in other species.

Response: Yes, the red light illumination causes an elongation of petioles and a downward turning of the leaves. This is a well-known phytochrome response that we cannot avoid in our system. It, however, has no critical influence on our approach since the induction is done in blue light and hits either cotyledons or the apical meristem, both face the light sources perfectly. Further the tissues are albino and the light penetration can be expected to be efficient even in unfavourable leaf positions.

For a transfer into another plant species we would first compare the *Arabidopsis* BLREs to those of the target species. If sequence differences can be identified we would isolate the corresponding sequences of the target species and implement them in our system.

While those limitations cannot be questioned, use of the system in *Arabidopsis* is sufficient to justify its utility. Furthermore the authors develop it for one specific purpose: to generate genetic stocks of mutants whose homozygosity is either lethal or, at least, incompatible with seed production, and therefore with propagation. This then allows refined experiments with homozygous material to identify the consequences of presence/absence of gene function, repeated multiple times: the relevant, homozygous stock is generated under standard growth (white light) conditions, and the control achieved by turning complementation off by transfer to monochromatic red (i.e. blue-less) light. This is entirely, beautifully achieved, applying the technique to study the function of the plastid-encoded RNA polymerase (PEP), through its requirement of nucleus-encoded PAP7/pTAC14. The study demonstrates the usefulness of the technique, and reports three key findings: the requirement for PEP function during a critical time window in cells in which early stages of chloroplast development are taking place, but very little later on, the fact that this requirement is (as expected) cell-autonomous, and lastly that PEP activity appears to be minimally dependent of photosynthetic function (photosynthetic electron transport) during these biogenesis stages. The unexpected result of

the activity of PEP under photosynthesis-inhibited conditions is precisely described by monitoring both transcript levels and overall protein composition of the relevant, dissected tissues. One issue may require further evidence to confirm the stated critical window and “point of no return” of PEP requirement. The evidence provided shows that seedlings grown for up to three days in red light (R), or in darkness, can have PAP7 activated by switch to blue (B) and the cotyledons green, but this is not the case after four days (of R or darkness). The authors conclude that PEP activation later cannot cause chloroplast greening. This interpretation would require demonstration that PEP activation by B did indeed take place after 4 days in R or darkness. The endogenous CHS activation system used might, conceivably, also have a window of competence, and this would explain the same observation. One easy test would be to monitor the transcript of the transgene, making use of the fused GFP, as done in Fig. 6.

Response: We thank the reviewer for this insightful suggestion which is a highly valuable experimental addition to our study. Actually, we have not thought about the possibility that the blue light induced CHS promoter system could have also a time window for competence. We thus followed the proposal of this reviewer and grew BVB09 seeds 2, 3, 4 or 5 days in either dark or R_{off} followed by a switch to B_{on} , respectively. We then measured the expression of the PAP7 construct in the cotyledons by qRT-PCR after 2 days of blue light illumination (a time point far before new leaves could emerge and compromise the experiment). We found a strong expression of the construct after the shift to B_{on} in all samples regardless of the tested pre-cultivation time. This indicates that the lack of chloroplast formation after 4 or more days of non-inducing pre-cultivation is not caused by a lack of responsiveness of the system. This observation is in favor of our hypothesis that intrinsic developmental constraints are responsible for the missing chloroplast biogenesis. The data were added as Supplementary Fig. 3g and the corresponding text in the results section was adapted accordingly.

The reviewer also found the cell-autonomy of the above response particularly intriguing. Thresholds of activity of essential, cell-autonomous processes in chloroplast biogenesis clearly exist, are probably operating in early proliferating cells in leaf primordia, and explain the development of cell sectors of contrasting phenotype, as seen in variegated leaves. The evidence for this cell-autonomous requirement is presented in Fig. 3 and shown by what seems a cotyledon (Fig. 3B, possibly derived from 3A).

Response: This is correct, Fig. 3B is a magnification of Fig. 3A. We mention that in the figure legends, but to make it more clear we connected now both images by lines and included a frame in Fig. 3A. In Supplementary Fig. 3 we added additional examples from other plants to demonstrate that the appearance of patchy cotyledons can be robustly reproduced.

Cotyledons are, of course, embryonic organs. Was such a phenomenon ever observed in newly-developing leaves emerging from the meristem? A seedling in Fig. 3F may show this, but the image is insufficient to confirm. It would be of value if the authors could clarify.

Response: The reviewer is absolutely right, cotyledons are embryonic tissues that may exhibit properties that deviate from follow-up true leaves. We, however, observed very similar phenomena in true leaves when we performed our light shift experiments presented in Fig. 2. For instance look to leaves 3a and 3b in Fig. 2e. These leaves emerged from the apical meristem at the end of the B_{on} condition and the leaf blade expanded later in R_{off} conditions. As a result you can see at the end of the photo time line (all photos show the same plant even if the orientation rotates) that the leaves 3a and 3b have green tips and edge while the leaf blades remain white with some light green patches. Such kind of mosaic patterns could be observed often in the shift experiments since

the BVB09 line grows faster in Bon than in Roff preventing a perfect match between induction times and emergence of leaf primordia. Actually we aimed to obtain perfect alternating green and white leaf pairs but realized that this would require extensive testing of variations in induction times to obtain perfect synchronicity (a project for the future). **We pointed towards leaves 3a and 3b in the results section and show additional follow-up leaves in Supplementary Fig. 3f. We further rephrased the discussion about the cell-autonomy and explain the phenomenon in more detail.**

Lastly, the authors display how transcript levels of a range of genes, nucleus- and plastid-encoded, are impacted by the absence of PAP7 or its complementation. Among plastid-encoded genes, those transcribed by the nucleus-encoded polymerase are (with the exception of rpoB) all presented in Fig. S4. Do the authors have an explanation for the low transcript levels of petD in the pap7 mutant and the R-grown BVB line?

Response: Because of the space constraints we selected only a few representative genes with general relevance for the study to appear in Fig. 6. This does not mean that the other genes included in the supplement are less important or less relevant. In fact, also these data include important results which are of great interest, however, more for a specialist rather than for a broad audience. **Therefore, we show and discuss only the most important data.**

The plastid petD gene belongs to the group of class II genes, thus it is transcribed by NEP and PEP. However, PEP is the dominant polymerase for its transcription and consequently petD transcript accumulation is very low in the homozygous pap7-1 control as well as in the BVB09 line grown in Roff. These expression data correspond well with those we reported earlier for petD (compare Grübler et al 2017 Plant Phys).

The text is generally well written, and the figures beautifully constructed to easily interpret. Note, however, that Fig. 2 is also labelled as a second Fig. 1. On several occasions Fig. 2 is referred to as Fig. 1, and viceversa.

Response: **Numbering has been corrected.**

Editing comments:

Repetitions > repeats. **done**

Line 79 put forward > advanced (to “put forward” is done only the first time) **done**

144 Should refer to Fig. 1B-C and Fig. S2C (lines BVB04m BVB09 and BVB12). **done**

154 No Fig. 1f exists. Fig. 2f does not show leaky expression of 4-repeat BVB

Response: Fig 1f was corrected to Fig 1e. Remark on Fig 2f likely refers to Fig. 1e. Leakiness in Chl production is poor but detectable and statistically significant when compared to pap7-1 as indicated by asterisks. It is however not well visible in the bar chart because of the high wt values. The precise values are provided with the Source Data sets now. Indeed, the leakiness is not reflected at the level of PAP7 expression, but **to avoid any potential ambiguity in further experiments we chose the 3-repeat BVB line that is tight in both parameters. This is discussed now in more detail as also requested by another reviewer.**

Fig. 6 What is given a value of 1, y axis?

Response: The expression data are given as „relative expression“ where the detected value refers to EF1a expression as internal control. **This is indicated in the figure legend now.**

261 Complicate > complex **done**

Fig. 1 legend Ron > Bon **done**

Methods should indicate the statistical tests used. Note that multiple testing increases errors, and there are ways to avoid those.

Response: The statistical tests have been mentioned now in the methods section and appropriate places in the manuscript.

Reviewer #3 (Remarks to the Author):

The manuscript by Uecker et al. addresses an important and timely question: the role of plastid-encoded RNA polymerase (PEP) in chloroplast biogenesis. Recent progress in structural understanding of PEP has only reinforced the importance of open questions regarding the biological roles of this crucial enzyme. Advancement in this field has been hindered by a challenging chicken-and-egg situation, where disruptions of PEP subunits impede chloroplast biogenesis, subsequently disrupting various processes, potentially including transcription itself. This manuscript addresses this conundrum by developing a novel inducible system, enabling precise control of PEP subunits (or any other proteins) through exposure to different wavelengths of light. Utilizing this tool, the manuscript investigates the role of PEP and elucidates two important aspects of its involvement in chloroplast biogenesis. (1) A limited time window for PEP-dependent chloroplast development, which is an interesting novel finding and (2) independence of chloroplast development of PEP transcription. Although these insights do not exhaust the power of the new inducible system, the current scope of the manuscript is appropriate and fully justified by the impact of the novel optogenetic system. This is a significant study that will establish a new standard for investigations of PEP subunits, other essential chloroplast proteins, and likely other essential plant proteins. The manuscript is well-written, technically solid and will be of interest to a broad audience of scientists studying genetics, not only in plants.

Specific comments

- There is no data showing behavior of the BVB optoswitch on the level of RNA or protein accumulation. It is important to know the level of induction and the level of leaky expression, which is a common problem of many inducible systems. Adding new experimental data providing these insights would substantially strengthen the manuscript.

Response: We gave expression data of the BVB gene construct in Fig. 1C and Fig. 3D as well as protein data in Fig. 6. These data are however endpoint data. We assume that with the term „behaviour“ the reviewer is asking for the properties of the system covering speed of responsiveness, maximum level of expression and the ability to shut off. These are indeed important properties to know especially in comparison to other inducible systems using chemical inductors. **To this end we performed a detailed kinetic experiment.** BVB09 and wt seeds were grown for 7 days in Roff conditions and PAP7 expression was determined by qRT-PCR. Subsequently, the seedlings were exposed to Bon for 24h to observe the induction followed by another shift to Roff to determine the shut-off properties. The induction of the BVB optoswitch reached wt levels within 30 min and maximum level after 4h with more than 2 times the expression level of wt. The shut-off was rapid (reaching less than 10% of max level within 1h and almost zero after 4h). With respect to responsiveness and tightness this system is definitely of high value. **We generated a new figure (Fig. 2h) and introduced these data into the manuscript right after the shift experiment data described in Fig. 2 including a description of the expression behaviour of the BVB optoswitch.**

- The mechanism responsible for the checkpoint that provides competence for chloroplast biogenesis should be discussed more thoroughly. For instance, is it possible that this checkpoint is mediated by nutrient availability in seeds?

Response: We expanded the discussion of this point as also requested by reviewer 1. Nutrient availability could be one possibility, but also other influences such as defined developmental checkpoints could be imagined. This question certainly will be a major focus of our future research.

- There is no justification of why the new optogenetic system is better than previously established inducible systems in plants. There is also no comparison to other optogenetic inducible systems.

Response: The reviewer is absolutely right, we did not describe and discuss the advantages of the optoswitch in comparison to other well established systems. The new figure describing the temporal and quantitative properties of the system, however, now provides a sound data base for such kind of discussion. Correspondingly, we included some remarks about speed, degree of responsiveness and leakiness as well as some more general consideration such as simplicity and suitability of the system in the discussion. Please see the revised manuscript for details.

- There is no discussion of potential impacts of growing plants in narrow spectra (red or blue light only) compared to white light, which is an important consideration for the interpretation of results obtained using the BVB optoswitch.

Response: Here, the reviewer raises another important point for the evaluation of the system. One could assume that the different light qualities generate side effects, and indeed in red light we see a clear petiole elongation, a well-known phytochrome response under continuous red light. However, we observed no major impacts of the narrow bandwidth lights on growth behaviour, seed production and photosynthetic capacity when compared to white light. This is likely caused by the adjusted light intensities (see Supplemental Fig 1). For the reaction center of PSII an equal number of red or blue photons will result in the same number of charge separations since the higher energy content of the blue photons is released as heat and only the energy content to excite P680 is used. From the viewpoint of photosynthesis the different light qualities are thus negligible. With respect to photomorphogenic responses, the different light qualities definitely have an impact which however is not relevant for the scientific questions we follow in our present study. We discuss this now in more detail and conclude that the BVB optoswitch is well suited for many different questions except if the studied process is affected by the light qualities (for example through the action of specific photoreceptors).

- DCMU treatment has been used in the past and citing and discussing original literature describing its limited impacts on chloroplast biogenesis would strengthen the conclusions of this manuscript.

Response: DCMU as photosynthesis inhibiting herbicide has been used in a plethora of studies, but mostly in analyses that study photosynthetic functions. DCMU effects targeting the process of chloroplast biogenesis were studied in only very few cases, and to the best of our knowledge the herbicide has been never used to elucidate a potential impact of photosynthesis on PEP formation. Doing an extensive literature search we identified some publications that deal with related aspects, but no one used DCMU in a way that corresponds to our specific experimental question. We, however, added two references describing DCMU effects on chloroplast biogenesis in the discussion that appear to us appropriate to meet the reviewers concerns.

- Impacts of DCMU on RNA accumulation are sometimes overinterpreted as impacts on gene expression. This part of the manuscript has limited influence on its conclusions and could easily be shortened.

Response: Actually the DCMU treatment aimed to block photosynthetic signals that were proposed to affect or regulate the build-up of the chloroplast RNA polymerase in early chloroplast biogenesis (see Diaz et al 2017 Nat. Commun.). The observed greening alone (Fig. 4) was not sufficient to prove this question. Therefore we did a micro-dissection of the greening tissues and checked the presence (by BN-PAGE, see Fig. 5) and the activity of PEP by determining the accumulation of plastid transcripts (Fig. 6). We detected only minor effects of the DCMU treatment on transcript accumulation indicating that photosynthetic electron flow has only very limited effects at this level, at least during the early stages of chloroplast biogenesis. While this appears as confirmatory evidence on first sight, it is however of high importance for the conclusions of our study as it demonstrates that the assembled PEP complex identified in the subsequent green leaves by BN-PAGE is indeed functional. Further, this figure provides additional information about efficiency and tightness of the BVB tool. It directly compares transcript accumulation in Roff conditions for BVB09 and pap7-1 indicating the tightness of the BVB tools (since we do not observe any increase in gene expression in BVB09 vs. Pap7-1). The figure further demonstrates the increase in expression in Bon conditions (including the increase after a shift) and finally provides also a comparison to WT under all these conditions. **We therefore would like to keep this figure as it is (in addition also to meet requests from reviewers 1 and 2), but we now include more cross-references in the manuscript to underline its importance and rephrased parts of the paragraph to make this more clear.**

- There are problems with labels of Fig 1F (referenced in the manuscript) and Fig 2 (mislabeled in figure legends).

Response: This has been corrected.

Point-by-point response

Reviewer 2

Minor corrections:

Line 405: Please replace “pretends” with “mimics”

Response: done as requested

Line 443 (and possibly elsewhere): Replace “a well known phytochrome response” with “a result to absence of cryptochrome and phototropin photoreceptors signalling”, or an equivalent expression. The current statement is incorrect. Phytochrome mutants, some in particular exhibit elongated petioles, therefore phytochrome signalling would shorten the petioles, and shade-light, therefore REDUCED phytochrome signalling, elongates them. The elongated petioles are a result of low cryptochrome action, and the down-turned or twisted laminae a consequence of absence of phototropin signalling.

Response: The reviewer is absolutely right, our statement was too cryptic and therefore misleading. We replaced it by a more detailed statement: „a result of absent cryptochrome and phototropin photoreceptors signalling due to the lack of blue light“.

Both changes are marked in yellow in the manuscript text.

We further added all other details (mostly statistical p-values) requested from your in-house editors as stated in the Author Check List. Please see further details in our responses added in that document.

In Figure 2 and Figure 5 we added the requested scale bars and molecular weight markers, respectively.